# A Siamese Neural Network for Behavioral Biometrics Authentication

## Abstract

The raise in popularity of web and mobile applications brings about a need of robust authentication systems. Although password authentication is the most popular authentication mechanism, it has several drawbacks. Behavioral Biometrics Authentication has emerged as a complementary risk-based authentication approach which aims at profiling users based on their interaction with computers/smartphones. In this work we propose a novel Siamese Neural Network to perform a few-shot verification of user's behavior. We develop our approach to authenticate either human-computer or human-smartphone interaction. For computer interaction our approach learns from mouse and keyboard dynamics, while for smartphone it learns from holding patterns and touch patterns. We show that our approach has a few-shot classification accuracy of up to 99.8% and 90.8% for mobile and web interactions, respectively. We also test our approach on a database that contains over 100K different web interactions collected in the wild.

## 1 Introduction

Biometric authentication has emerged as a complement to traditional authentication systems. The main advantage of such systems is that they rely on user's information that can not easily be stolen or crafted. Most active fields of biometric authentication in academia and industry are related to face authentication or fingerprint authentication, with a recent increase in interest on behavioral biometrics. Behavioral Biometrics authentication refers to the use of human-device interaction features to grant access to a specific service. This interaction could include, but is not limited to, typing patterns, mouse dynamics, smartphone holding patterns, voice recognition, gait recognition etc.

Machine learning algorithms have been proposed to verify users identity using behavioral biometrics features. Regarding behavioral biometrics in web environments (Human-Computer interaction), most of the work has focused on the use of Support Vector Machine and Random Forest classifiers to analyze mouse and keyboard interaction (Khan et al., 2018; Solano et al., 2020). Alternatively, some works have proposed to use built-in sensors available in mobile devices (i.e sensors information, touch interaction etc.) for authentication purposes (Rauen et al., 2018; Rocha et al., 2019; Zhang et al., 2016; Amini et al., 2018; Abuhamad et al., 2020). However, previous works in behavioral biometrics usually have three main drawbacks: (1) they need long interactions (minutes) in order to learn accurately the user behavior; (2) they require ad-hoc interaction challenges; or (3) they need a model per user to improve model accuracy.

In this paper, we present a Siamese One-Shot Neural Network (SOS-NN) which is able to assess a risk score after only one observation (i.e. enrollment behavior) of a given user. To achieve this, we propose a Siamese Neural Network architecture that assesses whether two behaviors belong to the same user. We present a similar architecture to user verification for both, web and mobile environments. In web environments, we create a set of features from raw mouse movements and keyboard strokes. On the other hand, for the mobile environment our SOS-NN analyses features created from touch interaction and motion sensors on the smartphone.

In sum, the contributions of our work are: (1) An approach to user authentication using behavioral biometrics information in an accurate few-shot learning fashion after only 5 seconds of user interaction; (2) A unified neural network architecture to authenticate user's behavior for both mobile and web environments that is able to achieve an accuracy of up to 99.8% and 90.8% respectively; (3) A framework which is able to accurately authenticate users in a large scale without requiring to

retrain the model for new users; (4) A systematic measurement study to understand the impact of the parameters to SOS-NN based on the authentication time window length and the $n$ number in the $n$-shot test; and (5) A comprehensive in the wild evaluation of our approach in web environments which test SOS-NN over thousands of users from real financial services.

## 2 BACKGROUND

In the field of biometric-based authentication, many sources of information have been proposed. These could be physical (facial recognition, fingerprint scanning, retina scan, etc) or behavioral patterns (signature verification, mouse dynamics, gait analysis, voice recognition, etc.). Behavioral biometrics has gained increased attention since reproducing the behavior of a legitimate user constitutes an unconventional challenge for attackers. Moreover, many behavioral biometric methods do not require specialized hardware, which makes a scalable deployment easier. While substantial advances have been achieved, there is still a gap to make such systems widely adopted in practice. We define as 'practical' methods that demand short periods of interaction per user (both for model training and for authentication), and simple architectures that ease deployment and maintenance. This paper proposes a novel approach, which complements traditional authentication in web and mobile environments, considering practical implementation characteristics and scalability constraints.

**Web Environment.** Multiple previous studies have employed multimodal biometrics in desktop environments to identify user's behaviour (Traore et al., 2012; Bailey et al., 2014; Fridman et al., 2015; Neha & Chatterjee, 2019; Solano et al., 2020). These studies propose to integrate, either at the feature or decision level, information from keyboard interaction, mouse dynamics and others. Most of these studies have evaluated classic ML classification models (e.g. SVM, Naive Bayes, Random forest and J48 algorithms). Few others (Jagadeesan & Hsiao, 2009; Khan et al., 2018), have explored the use of shallow neural networks. Remarkably, we found that siamese neural architectures had not been studied before in this field. Therefore, in this study, we explore the effectiveness, generalization and applicability of such architectures for behavioral biometrics.

Regarding user interaction, we highlight that, in multiple real-world applications there is a practical limit to the length of the interaction and amount of data that can be collected before deploying an authentication model, particularly when enrolling new users to the system. Therefore, we compared previous approaches on the amount of interaction required, per user, to train the model. We are comparable to few studies (4), that require between *2min.* to approximately *30min.* of user interaction to train the model. As an illustration, Khan et al. (2018) reported an accuracy of 97.3% using an SVM model per user, however, their approach would require recording at least 30 previous login attempts ($\approx$15 minutes) to train each user's model. A more recent approach proposed by Neha & Chatterjee (2019) achieves an accuracy of 95.6% after training a MLP for each user but they required 50 logins ($\approx$ 25 minutes) for training phase.

On the scalability perspective, previous studies use authentication paradigms that involve one model per user or multiclass classification, these methods translate into large infrastructure, deployment, monitoring and maintenance challenges. In contrast, our SOS-NN model generates a measure of similarity between two behaviors in a latent feature space. In this case, the question is not whether the sample belongs to a particular user, but rather if the samples are similar enough to conclude that the user is the same. This one-model-for-all (OfA) paradigm facilitates deployment and avoids further training for every new user registered in the system. To the best of our knowledge the SOS-NN model and OfA paradigm have not been applied in desktop/web behavioral biometrics before.

On Appendix A, we compare with further detail, previous approaches with respect to the classification methods used, the authentication paradigm and user interaction required.

**Mobile Environment.** Likewise, behavioral biometrics for authentication has also been implemented for mobile environments. Such models complement traditional authentication by taking advantage of the multiple built-in sensors available in mobile devices, being able to capture user behavior through several modalities. Some modalities rely on the use of mobile keyboard dynamics (Cilia & Inguanez, 2018); touchscreen interaction (Rauen et al., 2018; Rocha et al., 2019); or embedded motion sensors data (Abuhamad et al., 2020) to authenticate users. In order to strengthen security, especially against ad-hoc adversarial attacks, multimodal authentication frameworks have been proposed by researchers (Stanciu et al., 2016; Sitová et al., 2015; Lamiche et al., 2019; Acien

et al., 2019; Lin et al., 2018; Volaka et al., 2019; Shen et al., 2016; Centeno et al., 2018). Those methods rely on the fusion of multiple modalities of behavioral information (i.e. keyboard, sensors, touch, etc.) with the goal of having a better performance. Previous works, have achieved low False Aceptance Rate (FAR) and False Rejection Rates (FRR). However, they require a one-model-per-user paradigm (OpU) and long user interaction times (>*10min.*), which makes them challenging to use in real world scenarios. In our review, the best comparable performance was reported by Stanciu et al. (2016), with FAR and FRR equal to 0.14%, nonetheless, the system would require one K-Nearest Neighbors model per user and 20 previous logins for training each model.

Appendix A presents a detailed comparison between the main state-of-the-art studies developed in mobile behavioral biometrics in terms of the type of model requirements and performance characteristics. To the best of our knowledge we are the first study that uses Siamese Networks for mobile multimodal behavioral biometrics, with high resulting performance compared to existing solutions and capable of assessing a score after *5sec.* of user's interaction.

**Siamese Networks.** Siamese Neural Networks were first introduced by Bromley et al. (1994) to verify hand-written signatures. In general, Siamese Networks are composed of two twin sub-networks and a similarity module which compares the outputs of both sub-networks. Consequently, Siamese Networks are trained by feeding a pair of inputs which are processed by each twin in the network. Siamese networks have been used for verification tasks because of their capabilities to create embedding representations which minimizes similarity between samples from different classes (Melekhov et al., 2016; Boenninghoff et al., 2019). In the field of behavioral biometrics, Siamese Networks have been implemented to approach different behavioral modalities, like face recognition (Schroff et al., 2015; Taigman et al., 2014), signature recognition (Dey et al., 2017), gait recognition (Zhang et al., 2016), among others. Regarding the use of Siamese Networks for behavioral biometrics authentication in web and mobile environments, Centeno et al. (2018) used Siamese Networks along with CNNs as a tool to create embeddings from motion sensors plots, and then feed them into a one-class SVM classifier. More recently, Giot & Rocha (2019) proposed the use of Siamese Networks to approach static authentication model using keyboard dynamics in web environments.

Furthermore, Siamese Networks have been extensively used to approach classification problems in which the few samples of each class are available to learn from (i.e. *Few-shot Learning*) (Hindy et al., 2020). Particularly, it is possible to go as far as to limit the number of available samples to only one (*one-shot learning*). In one-shot learning literature, Siamese Networks have shown promising results in tackling classification tasks, under the restriction of observing only one sample before making a prediction over a test instance (Koch et al., 2015; Triantafillou et al., 2017). To the best of our knowledge there are no previous works on the use of Siamese Networks to approach multimodal behavioral biometrics authentication in web or mobile environments. Specifically, our approach differs from others in that it (1) focuses on one-shot learning, (2) implements semi-hard pair selection, and (3) learns from different behavioral sources in both web and mobile environment.

## 3 APPROACH

In this work we propose a deep learning framework to complement traditional authentication systems by analyzing behavioral biometric features. Such framework aims to learn and analyze inherent user behavior while interacting with a device in a few-shot fashion. In particular, we analyze two different environments to learn from users: *Web Environments* and *Mobile Environments*. For the Web Environment we are interested in authenticating users using information from mouse and keyboard dynamics. On the other hand, for the Mobile Environment we focus on the physical sensors like touch, accelerometer, gyroscope and magnetometer.

### 3.1 FEATURE ENGINEERING

As we are interested in behavior verification, the first step consists in processing continuous raw data sequences, recorded from the machine-user interaction, and then transform them into readable features for the model. Recorded sequences are split into multiple fixed length interaction windows for all the modalities recorded. In this paper we explored the performance of the model for multiple fixed-time windows varying from *5sec.* to *60sec.* of user interaction.

Table 1: Sets of behavioral features for keyboard interactions in Web environments.

| Key Down-Up | Key Up-Up | Key Down-Down | Key Up-Down |
|---|---|---|---|
| Length of time from a key is pressed until it is released. | Latency from one key is released until the next key is released. | The latency between two consecutive keys pressed. | Length of time from one key is released until the next key is pressed. |

**Web Environment.** For the Web Environment, we start out from the raw sequences of mouse movements and key presses. The raw mouse data includes timestamp, pointer coordinates $(x, y)$ and the type of interaction (i.e. click, mouse movement, etc.). We transform the raw data by designing two sets of features, inspired in Ahmed & Traore (2007); Solano et al. (2019). To calculate features that capture the direction of the mouse moment, the angular space is split into eight equal bins of $45°$ and each mouse event in the fixed-time window is classified into one bin. Then, the first set of features is calculated by finding the average movement speed in each of those direction partitions. The second set is the proportion of movements performed in each direction along the fixed-time window. Mouse dynamics were captured from 16 features (2 sets $\times$ 8 directions). In a like manner, the keyboard raw data includes the timestamp, key and key interaction (i.e. Press or release). Four sets of keyboard features were created as described in Table 1. Afterwards, for each set of keyboard features, we calculate the average, the median and the standard deviation. As a result, for the keyboard dynamics information we build a set of 12 features (4 sets $\times$ 3 metrics).

**Mobile Environment.** For the Mobile Environment, the raw data is represented by (1) measurements from sensors (gyroscope, magnetometer and accelerometer) and (2) touch inputs performed by the user along the fixed-time window. Regarding the sensor's measurements, we record values in $X, Y$ and $Z$ axis for each sensor. For each sequence, including sensors and touch data, we compute 5 measures of central tendency of the data distribution: mean, standard deviation, median, minimum and maximum. On the other hand, for the touch interaction we record raw data from the touch's center, the touch's pressure and the touch's size. From these records we compute 4 features: mean touch duration, the average number of changes in pressure or touch center within the same touch (down-up) interaction, standard deviation of touch's center $(x, y)$ and mean finger size (touch size area). We added two features related to mobile keyboard interaction, based on latency between consecutive touches. Altogether, for each fixed-time window we have a vector containing 52 features (45 sensors + 5 touch + 2 keyboard).

### 3.2 SIAMESE NEURAL NETWORK

Our Deep Learning Framework evaluates if two recorded behaviors belong to the same user. This approach could authenticate a user after only one observation (i.e. the enrollment behavior). Our SOS-NN computes the similarity between two behavior inputs. In that sense, if two inputs are similar enough, our system concludes that the incoming behavior belongs to the legitimate user. The siamese network architecture is made of two identical sub-networks; In our case, two fully connected neural networks which share weights. Each sub-network processes one of the input behaviors and works as a feature extractor.

Both of the sub-network's outputs are bounded by an energy function. We compute a $L_1$ *distance* as the energy function between both of the computed feature vectors in the latent space. Intuitively, this distance should be large when input behaviors belong to different users but small when they belong to the same user. Following the energy function calculation, we include in our model a fully-connected decision network, which makes the classification decision based on the distance between the feature vectors in the latent space. Consequently, the output of our SOS-NN is a binary classification, where the output is `One` if behaviors belong to the same user, and `Zero` otherwise. Figure 1 depicts the proposed SOS-NN architecture.

### 3.2.1 SAMPLE GENERATION

The Siamese networks learn from comparing pairs of behaviors. A positive pair is defined as a pair of two behaviors which belong to the same user, whereas a pair of behaviors from different users is labeled as a negative pair. The model is trained by presenting multiple samples of these

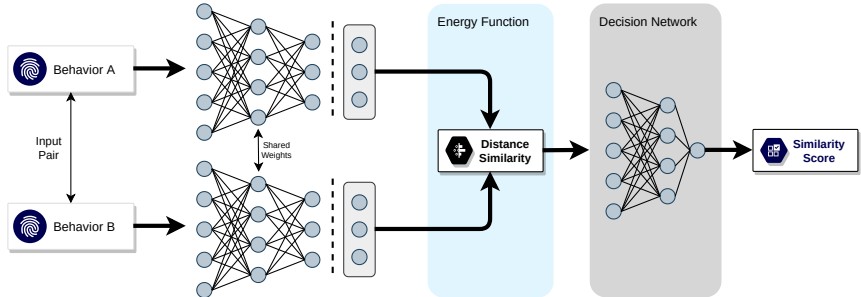

Figure 1: SOS-NN architecture.

pairs, the training goal is to minimize the energy function (i.e. $L_1$ *distance*) between behaviors in positive pairs, and to maximize the energy function for negative pairs. The quality of retrieved pairs in training will determine the quality of our SOS-NN. The naive selection of those positive and negative pairs is a random selection. This naive approach is fine for the positive pairs but for the negative samples it is crucial to select high quality pairs. More advanced techniques can be used to select the training samples, like triplet loss technique, which uses simultaneously 3 examples to optimize every training step (Hermans et al., 2017). The constrains on the distances between the triplets can be controlled; we chose Semi-hard negatives as our strategy to populate triplet samples. Further details on the triplet loss selection can be found on Appendix B. On the evaluation we compare two pair generation strategies to feed with the Siamese Networks, namely (1) Naive Pair and (2) Semi-hard Triplets.

### 3.2.2 TRAINING STRATEGIES

The training procedure depends on the sample generation strategy (naive or triplet). For the naive pair strategy, we train our SOS-NN using a cross entropy objective loss on our binary classifier (same or different user). Therefore, for the naive strategy, we perform the weights optimization over the full network (including both the siamese and decision networks), using standard back propagation. Notice that gradient is additive over the tied sub-networks. On the other hand, for the triplet strategy the training is done in two steps: (1) siamese network training and (2) decision layer training. To train the siamese network, we use the Semi-Hard Triplets configuration. Once the feature extractor is trained, their weights are frozen and the fully connected decision layer is appended to complete the SOS-NN. Lastly, we train the decision layer by using binary cross-entropy as a loss function.

## 4 EVALUATION

### 4.1 EXPERIMENTAL SETUP

**Web Environment Dataset**. We collected two datasets under *controlled* and *uncontrolled* conditions. For the *controlled setting*, we used data acquired through the *'Amazon Mechanical Turk' service*[1] by submitting the task of logging on a website designed to capture mouse and keyboard events. Here we collected interactions from 89 worldwide workers who introduced fictitious credentials on the login website. A total of 1374 sessions (full login interaction) were obtained with an average duration of *26.7sec.* per session. As for the *uncontrolled setting*, a monitoring system was appended to two login web pages from real banking domains where mouse and keyboard events were recorded. To protect users privacy, the raw data was transformed into features on the client-side before they were sent to the server. In this setting, 807622 sessions worth of interactions were collected from 125403 users, with an average of *22.4sec.* per session.

**Mobile Environment Dataset**. We developed a realistic looking Android application simulating different banking activities and equipped with a event logger to save information related to touch, accelerometer, magnetometer and gyroscope events. For keyboard touch events the timestamp and key value were logged. We collected data from 35 volunteers performing sessions lasting *10min.* on

---

[1]A service where human workers perform a certain task following instructions defined by the task requester.

average and making up to *372min.* in total. Moreover, the volunteers used more than 20 different smartphone devices. In case of acceptance both datasets will be made publicly available.

**Feature Engineering**. The behavioral data gathered in controlled settings was subsequently merged into a continuous session for each user. Afterwards, the full history per user was split into fixed-time windows. We analyzed time windows of 5, 10, 20, 30 and 60 seconds of interaction. Next, for each fixed-time window we calculate the features following the methodology described in Section 3.1. Regarding the data collected in the wild from real banking domains, since features are pre-computed on the client-side, we collected login sessions with arbitrary durations. Finally, we randomly split the dataset by users, since we wanted to avoid validating with behaviors similar to the ones observed in training. Splitting the dataset by users has been shown to perform better than other train-test schemes in previous verification tasks (Koch et al., 2015). More details related to data processing and train-test splits can be found in Appendix C.1.1.

**Networks Configuration**. We implemented our approach using the Tensorflow framework. The feature extraction and classification decision is made by a siamese network and decision network respectively, both were built with fully connected architectures. Therefore, regularization, batch normalization and dropouts were tested for both networks. For the triplet loss we used the `SemiHardTriplets` implementation available on Tensorflow Addons (Hermans et al., 2017). Our SOS-NN network is trained using Adam optimizer. For more details refer to Appendix C.2.

## 4.2 RESULTS

To understand the capabilities and robustness of our proposed SOS-NN we perform a systematic evaluation over the model performance for different setups in the training and verification phase. Variables included in this systematic evaluation are the sample generation scheme (pair o triplet), interaction time required (fixed-time window length) and the number of samples (*n*) from previous history to compare with (*n-shot* testing). For the n-shot testing, the output consists in the average of each individual pair-comparison. At the end of the section, we show an evaluation of our SOS-NN for web environments over a large scale experiment (≈125 K of users) to verify that performance remains at the same level under uncontrolled conditions.

**Web Environment Results.** Figure 2a illustrates the model accuracy in web environments for different configurations. As can be noticed from Figure 2a, the verification accuracy is consistently higher for longer time windows in the verification phase. The best verification accuracy obtained for web environment is 90.8%. Moreover, we also found a gradual rise in the verification accuracy when the number of comparisons increases. In the *1-shot* and *5-shot* verification tasks our approach achieves up to 74% and 86% in verification accuracy respectively. Table 2 summarizes the accuracies for different verification windows requirements with higher values in darker color. As we expected, the performance of our model is better when analyzing longer times in verification phase. However, more comparisons seems to have a larger effect on accuracy than longer interactions times. For instance, 10 comparisons (*10-shot*) for window lengths of 5 seconds is better than one comparison (*1-shot*) of 60 seconds, as can be observed in Table 2b. We believe this is due to multiple comparisons smoothing the score of the incoming fixed-time windows.

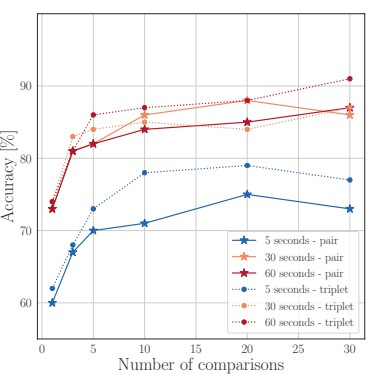

(a) Web Environment

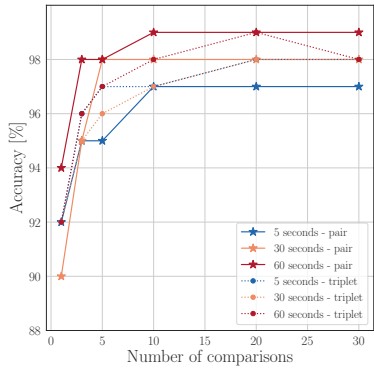

(b) Mobile Environment

Figure 2: Accuracy results for evaluation over several n-shot comparisons and time windows interactions for behavior verification collected in web and mobile environment.

Moreover, we observe an improvement of around 4% when using triplet setup for training phase. This improvement is a consequence of better negative sample choices in the training phase, which leads to a more powerful feature extractor at the bottom of our SOS-NN. Recall that in the triplet training scheme training samples are forced to satisfy Equation 1. Consequently, our results confirm that a better choice of behavioral positive and negative samples could moderately improve the performance of our SOS-NN for web environments. Furthermore, the verification accuracy reaches the best values after 10 comparisons (*10-shot*) as it starts fluctuating close to the maximum obtained performance. Figure 3 shows the ROC curve and the FAR and FRR for the best setup in web environment. As can be observed from Figure 3b, the FAR and FRR are 9.8% and 16.2% respectively for classification threshold equal to $0.45$.

**Mobile Environment Results.** Results for the mobile environment can be found on Figure 2b. The best verification accuracy obtained is 99.8%. We found a considerable accuracy improvement (5%) when the number of $n$ comparisons increases from 1 to 10 comparisons. However, from 10 comparisons until 30, for all window lengths the accuracy is saturated around 98% for a *60sec.* window and 97% for a *5sec.* window. Besides, a 1% difference between a 5 and a 60 seconds-window suggests that the information collected in the first seconds gives already a satisfactory description of user behavior. The difference between pair and triplet training is not significant. This suggests for high validation accuracy (i.e. >90%), the performance depends more on the architecture of the sub-networks or on data structure aspects instead of the sample generation strategy.

Finally, if we focus on an ideal setting where there are no restrictions on how much data can be sampled from each user, meaning no limit on comparisons or windows lengths, we achieve a validation accuracy up to 98% for all with windows above 20 comparisons (See Table 3). Nevertheless, in the case of a practical solution (i.e a company's product) there are some limitations to be taken into account. Here, waiting too much to gather several 60-second windows can compromise the security of the system. In this case, the lesser the time the better: even for the case of a 5-second window and only one comparison, namely one-shot inference, we achieve an accuracy of 92%. Our preferred case would be using 3 comparisons, which corresponds to *15sec.* of interaction, easily collected after only one login session of the user; in this scenario our accuracy is 96%. Figure 3 shows the ROC curve and the FAR and FRR for the best setup in mobile environment. The best FAR and FRR are 0.01% and 0.22% respectively for threshold classification equal to $0.4$. Remarkably, our SOS-NN achieves an AUC of 0.99 in mobile environment (See Figure 3a).

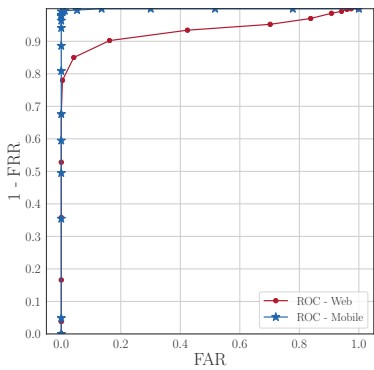

(a) Receiver Operative Curve (ROC).

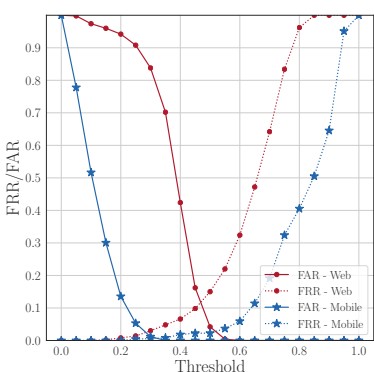

(b) FAR and FRR for all classification thresholds.

Figure 3: Performance of the model for the best setup (*20sec.*) in web and (*5sec.*) mobile environment.

Table 2: Accuracies for test in Web Environment.

(a) Accuracies for pair training.

| | | Time Length [sec] | | | | |
|---|---|---|---|---|---|---|
| | | 5 | 10 | 20 | 30 | 60 |
| Comparisons | 1 | 0.60 | 0.64 | 0.67 | 0.73 | 0.73 |
| | 3 | 0.67 | 0.71 | 0.74 | 0.81 | 0.81 |
| | 5 | 0.70 | 0.75 | 0.78 | 0.82 | 0.82 |
| | 10 | 0.71 | 0.78 | 0.79 | 0.86 | 0.84 |
| | 20 | 0.75 | 0.80 | 0.81 | 0.88 | 0.85 |
| | 30 | 0.73 | 0.80 | 0.82 | 0.86 | 0.87 |

(b) Accuracies for triplet training.

| | | Time Length [sec] | | | | |
|---|---|---|---|---|---|---|
| | | 5 | 10 | 20 | 30 | 60 |
| Comparisons | 1 | 0.62 | 0.66 | 0.71 | 0.74 | 0.74 |
| | 3 | 0.68 | 0.75 | 0.80 | 0.83 | 0.81 |
| | 5 | 0.73 | 0.74 | 0.79 | 0.84 | 0.86 |
| | 10 | 0.78 | 0.79 | 0.85 | 0.85 | 0.87 |
| | 20 | 0.79 | 0.80 | 0.88 | 0.84 | 0.88 |
| | 30 | 0.77 | 0.81 | 0.86 | 0.87 | 0.91 |

### 4.2.1 EVALUATION IN THE WILD

In order reach a deep understanding on how our SOS-NN behaves in the wild at a large scale, we tested the framework with over 100 thousand real users from legitimate banking domains. Figure 4 depicts the verification accuracy of our approach evaluated in real web sessions. First of all, we simulated the same setting we had in the controlled experiment: we trained with 50 users and tested with the remaining users. In this baseline, the verification accuracy for *1-shot* testing is 63% and 68% for pair and triplet training respectively. Notice that this performance value is very close to metrics we had for the controlled dataset for a time window length of 10 seconds. Recall that 75% of sessions in the uncontrolled dataset lasted less than 10 seconds. In like manner, the accuracy when testing in a *5-shot* and a *10-shot* fashion is up to 75% and 79% respectively.

More importantly, we investigated how the model performance increases when more data is considered to train it. Accordingly, we study the verification accuracy or our SOS-NN when behavioral data from 25K, 75K or 100K different users is included in the training set. Figure 4 shows the verification accuracy for different *n-shot* configurations when different amounts of users in training data. From Figure 4 we can observe that model performance jumps when more user's behaviors are in the training phase. In general, the rise in verification accuracy when training with 25K in comparison with 50 users is about 9%. In addition, we do not observe sharp improvements when training with more than 25K users. We believe this is happening because the behavior of 25K users is complex enough to force the network to learn the differences in the latent space among a diverse spectrum of user behaviors. This finding suggests that, for behavioral biometric data, there could be a limit to the representational power of the data in the latent space, this is an interesting insight for future research on siamese neural networks applied to structured data.

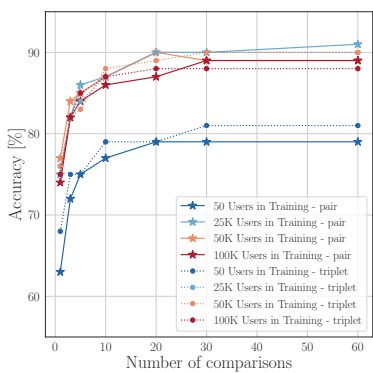

Figure 4: Accuracy results for evaluation over several n-shot comparisons for web behaviors collected in the wild (125K users).

### 4.2.2 DISCUSSION

As shown, our SOS-NN is able to accurately classify behaviors for short and long interaction windows, and therefore it is suitable as a competitive, maintainable and lightweight mechanism for authentication in contrast to previous work (Appendix A). Previous schemes are mainly based on either one model per user or a multiclass classification, which could result in a resource intensive implementation in the real world and arguably not as scalable for thousands or even millions of users as our proposed one-for-all approach. In fact, for multiclass classification paradigm the model has to be re-trained every time a new user is added; while for the one-model-per-user a model has to be trained for new user in the service (requiring high volumes of data for each new user). By comparison, after training with a certain quantity of data our SOS-NN is ready to be deployed and used with any user, even if it has not seen before by the system.

Additionally, we studied the system performance in a real authentication scenario, where requiring short interaction windows and rapidly deploy a model that can assess a risk score for the user can be a large advantage. The majority of examples in the related work require minutes or even hours of interactions or tenths of logins to achieve their performance. Alternatively, our approach could

Table 3: Accuracies for test in Mobile Environment.

(a) Accuracies for pair training.

| | | Time Length [sec] | | | | |
|---|---|---|---|---|---|---|
| | | 5 | 10 | 20 | 30 | 60 |
| Comparisons | 1 | 0.92 | 0.90 | 0.93 | 0.90 | 0.94 |
| | 3 | 0.95 | 0.95 | 0.94 | 0.95 | 0.98 |
| | 5 | 0.95 | 0.95 | 0.97 | 0.98 | 0.98 |
| | 10 | 0.97 | 0.96 | 0.98 | 0.98 | 0.99 |
| | 20 | 0.97 | 0.96 | 0.99 | 0.98 | 0.99 |
| | 30 | 0.97 | 0.97 | 0.99 | 0.98 | 0.99 |

(b) Accuracies for triplet training.

| | | Time Length [sec] | | | | |
|---|---|---|---|---|---|---|
| | | 5 | 10 | 20 | 30 | 60 |
| Comparisons | 1 | 0.92 | 0.92 | 0.91 | 0.92 | 0.92 |
| | 3 | 0.96 | 0.96 | 0.94 | 0.95 | 0.96 |
| | 5 | 0.97 | 0.97 | 0.96 | 0.96 | 0.97 |
| | 10 | 0.97 | 0.98 | 0.97 | 0.97 | 0.98 |
| | 20 | 0.98 | 0.98 | 0.98 | 0.98 | 0.99 |
| | 30 | 0.98 | 0.99 | 0.98 | 0.98 | 0.98 |

authenticate the user after only 25 seconds to 3 minutes with an accuracy of 73% to 85% in Web environment. We understand this value could seem relatively low, but that is because we push to the limit the capabilities of the model to learn behaviors even from short interactions as a trade-off between accuracy and promptness of assessment.

Finally, the difference of performance between web and mobile environments is noteworthy. The complete framework behaves way better with mobile data, even when the architecture and training procedure are the same. Recall that for the mobile environment a few seconds of interaction are enough for the model to achieve 99% of accuracy. This can be explained given that our model receives the devices' sensors input even when the user is not actively interacting with the application and is only holding the device. On the other hand, the web environment depends on the user interacting with the peripherals (mouse & keyboard), otherwise the system goes blind due to insufficient input data to make a valuable prediction, as seen in Figure 2a. This means that our web system needs more data, and therefore more time to be able to complement the authentication, but nonetheless can achieve reasonable performances of up to 90% of accuracy.

## 5 CONCLUSION

In this paper we have presented an effective approach to generate a Siamese Neural Network model that integrates different human-device interaction sources (mouse, keyboard or mobile sensors) for behavioral biometric authentication. Our model obtained high accuracy even when tested in a few-shot fashion, that is, needing only a few behavior samples per user. We also showed that the proposed model architecture can be easily adapted to both web and mobile environments. Furthermore, our system has the potential to rapidly scale in production environments because it needs only one model to evaluate behavior of many users. We evaluated our approach on various datasets, including production data for thousands of users in a web-application. In future work we plan to evaluate our approach on production data belonging to mobile users as well.

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

APPENDIX

# A  RELATED WORK

We collected information from previous studies for both desktop/web and mobile environments. The following table compares these works in terms of model inputs, model type, model performance, authentication paradigm, required amount of user interaction to train the model and required amount of interaction to authenticate the user, once the model has been trained.

Table 4: Related Work - Web and Mobile Behavioral Biometric Authentication

| | Reference | Input | Method | Paradigm | Performance | Training (Tr) & Authentication (Auth) Interaction per User |
|---|---|---|---|---|---|---|
| **Web** | Jagadeesan & Hsiao (2009) | K, M | KNN, ShNN | OpU | Acc. 82.2% | Tr:20 Sessions Auth:1 Session |
| | Traore et al. (2012) | K, M | BN | OpU | EER 8.21% | Tr: 200KS,2500ME Auth:1 session |
| | Bailey et al. (2014) | K, M, GUI | SVM, BN, J48 | MC | FAR 2.24% FRR 2.10% | Tr: 12h Auth: 10m |
| | Fridman et al. (2015) | K, M, St | NB, SVM | MC | FAR 0.004% FRR 0.01% | Tr: 33.6h Auth: 30s |
| | Khan et al. (2018) | K, M | SVM, MP, AB | OpU | Acc. 97.3% | Tr: 30 logins Auth: 1 login |
| | Neha & Chatterjee (2019) | K, M | J48, BN | MC | Acc. 95.6% | Tr: 50 logins Auth: 1 login |
| | Solano et al. (2020) | K, M | RF | OpU | FAR 23.34% FRR 10.73% | Tr: 2min Auth: 30s |
| | **Ours** | K, M | SOS-NN | OfA | FAR 3.62% FRR 27.3% Acc. 88% | Tr:6.6min Auth:20s |
| **Mobile** | Sitová et al. (2015) | K, TI | PCA SVM | OpU | EER 7.16 % | Tr:10min Auth:20s |
| | Shen et al. (2016) | TI, MS | O-SVM | OpU | FRR 6.85% FAR 5.01% | Tr:2700 logins Auth:3s |
| | Stanciu et al. (2016) | K, MS | KNN | OpU | EER 0.14% | Tr:20 logins Auth:1 login |
| | Centeno et al. (2018) | MS | CNN O-SVM | OpU | Acc 95.8% | Tr:270s Auth:2s |
| | Lamiche et al. (2019) | MS, K | MP | OpU | FAR 0.68% FRR 7% | Tr: 5 trials (WK + KS) Auth:1 trial |
| | Volaka et al. (2019) | MS, TI | MP | OpU | EER 15% | Tr:≈ 80min Auth: ≈ 9min |
| | Acien et al. (2019) | MS, AppU | SVM per-source | OpU | EER 15% | Tr:224 SS($\approx$ 9h) Auth:1 SS($\approx$ 224s) |
| | **Ours** | MS, K, TI | SOS-NN | OfA | FAR 0.01% FRR 0.22% Acc. 96% | Tr: 15s Auth: 5s |

**Input:** K: Keyboard Interaction, M: Mouse Dynamics, MS-Motion Sensors, TI-Touchscreen Interaction, GUI:Gui Features, St:Stylometry Features, AppU: App-usage time.
**Method:** KNN: K-Nearest Neighbors, ShNN: Shallow Neural Network, BN: Bayesian Network, NB: Naive Bayes, RF:Random-Forest Classifier, MP: Multi-Layer Perceptron, AB: Adaptative Boosting, O-SVM: One-class SVM, Siamese One-Shot Neural Network (SOS-NN).
**Paradigm:** OpU: One Model per User, MC: Multi-Class Classification, OfA: One Model for All.
**Performance:** EER: Equal Error Rate, FAR: False Acceptance Rate, FRR: False Rejection Rate, Acc.: Accuracy.
**Tr/Auth Interaction:** ME: Mouse Events, KS: Key strokes.

## B  FEATURE ENGINEERING DETAILS

### B.1  WEB ENVIRONMENT

The mouse features used in our web environment approach are inspired in Ahmed & Traore (2007); Solano et al. (2019). These features build a mouse dynamics profile over the movements performed by a user in a fixed-time window. The key idea is to analyze mouse actions like mouse movements, drag-and-drops and clicks within eight equal area partitions in the circular space surrounding the pointer's position at any given time along a recorded session. We perform this analysis by getting the next pointer's position and calculating the movement direction, the speed, the angle of curvature and the curvature distance (See Figure 5). In consequence, a unique user signature can be built by calculating central tendency statistics for each partition.

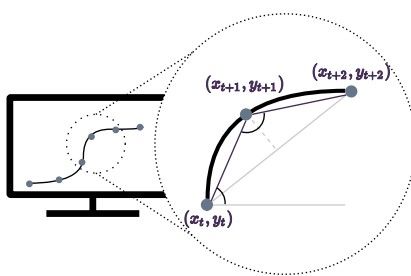

Figure 5: Features engineering illustration for web environment.

As for the keyboard, the timing features are extracted from the timestamps and the type of action (*key-down* or *key-up*) of the keyboard events recorded. The aim here is to discover unique user tendencies by measuring how long was a key pressed (*down-up*), the time between two consecutive key releases (*up-up*), the time between two consecutive keys pressed (*down-down*) and the time from one key release until the next key is presses (*up-down*). Thus, the keyboard features consist of the central tendency statistics per each of these metrics.

### B.2  MOBILE ENVIRONMENT

The mobile features are intended to capture the way a user holds the smartphone and simultaneously their touch pattern when tapping the screen. In that sense, we start from the raw measurements from smartphone's sensors (i.e. gyroscope, magnetometer and accelerometer). Notice that the value of those sensors change constantly while the user naturally interacts with the smartphone. In sum, we calculate central tendency statistics over the series $S_1, S_2, ..., S_{tn}$ for each sensor $S \in [gyroscope, magnetometer, accelerometer]$. Then we build the user device holding profile by analyzing the 45 sensor features (3 Sensors x 3 Axes x 5 statistics). Regarding touch interaction, we use the interaction derived from touch actions as the raw data. Afterwards, we compute the mean touch duration, the average number of changes in pressure or touch center within the same touch (down-up) interaction, standard deviation of touch's center (x, y) and mean finger size (touch size area) over all in actions in fixed-time window.

In summary, the features used to capture meaningful traits of user behavior when interacting in web and mobile environments are summarized by input source in Figure 6.

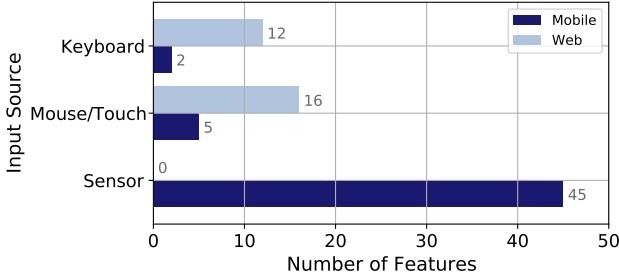

Figure 6: Count of behavioral features per sources for web and mobile environment.

## C  TRAINING DETAILS

### C.1  SAMPLE GENERATION DETAILS

One of the most successful approaches applied to Siamese Networks in order to improve the selection of negative pairs was presented by Schroff et al. (2015). As shown early, a Siamese Networks is an architecture that compares two sets of inputs; however, Schroff et al. (2015) implemented a triplet technique that uses 3 examples at the same time to optimize every training step. These 3 examples are carefully selected so that the first two of them belong to a positive match (i.e. two examples that correspond to the same class, namely an anchor *(A)* and a positive example *(P)*) and the last one corresponds to a different class, being a negative example *(N)*. The goal is to minimize the distance in the feature space between examples of the same class and maximize the distance between examples of different classes, which can be expressed as:

$$|f(x_A) - f(x_P)|^2 + \alpha < |f(x_A) - f(x_N)|^2 \tag{1}$$

Where $\alpha$ is a margin parameter related to the difficulty of the examples. The final loss can be represented as:

$$L(A, P, N) = max(|f(x_A) - f(x_P)|^2 + \alpha - |f(x_A) - f(x_N)|^2, 0) \tag{2}$$

As in the pair generation, the 3 examples per triplet can be generated randomly as long as they fulfill the previously named condition. Nevertheless, this strategy would deliver most of the time a small Loss, because it is expected that many negative examples produce features with larger distances from the anchors, whose loss ends up adding nothing to the training. A better approach is to pre-select the triplets in order to optimize the learning. This pre-selection could be performed by choosing two kinds of triplets: Hard negatives and Semi-hard negatives. Hard negatives are examples whose feature distances with the anchor are strictly smaller than the distance between the positive example and the anchor (Schroff et al., 2015). Semi-hard negatives are examples whose distance with the Anchor is greater than the positive example, but still inside the margin $\alpha$ (Hermans et al., 2017). In principle, the hard negatives deliver the greatest losses, and therefore the strongest convergence, but in practice they can be too aggressive and collapse the loss function. Consequently, we use Semi-hard negatives as our strategy to populate triplets in training phase. With this in mind, we propose and compare both pair generations strategies (Naive Pair and Semi-hard Triplets) to feed with the Siamese Networks.

### C.1.1  DATA PROCESSING

Once the features have been calculated, we split the dataset into three disjoint set of users: 64% of the users for training, 16% for validation and 20% for test. Finally, the features were transformed in order to follow a normal distribution by using a non-linear transformation on each feature independently (Quantile Transformation in Pedregosa et al. (2011)).
**Privacy Concerns.** Collecting behavior sequences could lead to privacy concerns as personal data like passwords patterns (keyboard Dynamics), user location (sensor Analysis), etc. is being recorded. To ensure privacy of the information, we transform the raw sequences in the client-side and send the behavioral biometrics features to be analyzed by our SOS-NN model in the server-side.

### C.2  NETWORK DETAILS

One of our goals was to implement a system that was almost the same for both web and mobile environments. The biggest difference is found at the input of the Siamese Networks, because the number of crafted features is different for each environment, namely 28 for web and 52 for mobile. After that, both environments share the same architecture and hyper-parameters. The feature extractor was composed by a fully connected network; the output of both Siamese extractors was compared with an $L_1$ distance layer.

The feature extractor was composed by a couple of dense layers with *ReLU* activation and a last layer with a *Sigmoid* activation. For the decision module, another fully connected network was

implemented whose input was the $L_1$ distance layer between the features in the latent space. We also tested other distances as energy function, like $L_2$, *Manhattan* and *Cosine* distances as well, but $L_1$ showed the best results. The decision subnetwork was composed of two dense layers with *ReLU* activation and finally an output network with a *sigmoid* activation, whose output can be understood as the probability of two inputs samples to come from the same user. We also tried different strategies to improve the validation accuracy like regularization, batch normalization, a deeper and wider network but the best validation performance was obtained with the simplest architecture.

The best parameter configuration was a mini batch of 100, a learning rate of 0.001, an Adam Optimizer and 200 steps per epoch. Weights were initialized following a normal distribution with zero-mean and standard deviation 0.01. For the feature extraction training with triplet configuration, the training was performed up to 1000 epochs, with an early stopping callback focused on validation loss. The training of this phase includes a $L_2$ distance as part of the available in the Tensorflow implementation of triplet loss schema ($\alpha = 1.0$). It should be clear that $L_2$ was used only for the feature extraction training; later when it is appended to the decision layer, a $L_1$ distance is used to measure the difference between two inputs. The decision layer subnetwork for the triplet approach was trained for 20 epochs. On the other hand, the training for pair approach, that is the feature extractor and decision layer as a whole were trained for 20 epochs. The data selection for the training is the one explained in Section 3.2.2.

