# OpenReview forum: "A Siamese Neural Network for Behavioral Biometrics Authentication"
_ICLR.cc/2021/Conference — Reject_

### Official Review · AnonReviewer3 · 2020-10-28
**Good article, but most suitable as a workshop submission**

**Rating:** 5
**Confidence:** 4

**Review:**

Summary: This paper proposes the use of siamese neural networks for the task of user verification by learning to identify their behavioral patterns of interaction with mobile phones or web browsers. Notably, the behavioral characteristics of users when engaging with these devices, measured in the form of mouse movements or keyboard typing patterns on browsers or touch patterns on mobile phones, serve as a proxy for more direct biomarkers such as face recognition, eye scans, etc. The authors argue that the latter requires more specialised hardware and is thus not easily scalable. Instead, they architect the first NN based solution that serves both modalities, web and mobile phones, while trying to be more accurate and responsive.

Review:
(a) The study is of good quality. The language is clear and concise. Related work is well covered to the best of my knowledge, making reference to one of the oldest instantiations of such model for signature verification.

(b) The originality of the method viz-a-viz the domain 'of biometric verification' seems to be high. In particular, existing solutions to the problem of behavioral biometric verification have either used different models (SVMs or RDTs) or different features or different formulations (classic supervised recognition setting using one model per user). This disposes them to problems such as limited scalability, lower performance and high latency. Herein, their solution of using a single siamese network, for both modalities, and across different users, adds generality, performance and speed. They have curated experiments with different number of users at training time, and using different metric losses (pairwise and triplet) for representation learning via similarity matching, shedding light on the effectiveness of their approach.

(c) That said, originality in terms of research contribution is limited. Siamese NNs have long been in use and recently revisited for variety of vision tasks such as face recognition[1, 2], object tracking [3], etc. The loss formulation used here is also standard, not going beyond the staple pairwise and triplet. Papers where newer losses or network architectures or training paradigms have led to better analysis and results have made sound research contributions and exhibited suitability for the main track e.g. [1, 2]. In other cases, where tasks are key, and results are good but the approach is largely borrowed, it serves good to go for a workshop track e.g. [3].

(d) Comparisons with existing methods in the domain have not been included to the main paper (they are supposed to be in the appendix). The aim perhaps of this paper is to present a baseline NN-based approach for user verification/ biometrics. I believe this would be a good fit for presentation at a more focused workshop on biometric verification. As a main track ICLR submission I find it lacking in novelty, reach and research contribution.

[1] Triplet Loss in Siamese Network for Object Tracking, Dong and Shen (ECCV 2018)
[2] Occlusion Robust Face Recognition Based on Mask Learning With Pairwise Differential Siamese Network, Song et.al. (ICCV 2019)
[3] Fully-Convolutional Siamese Networks for Object Tracking, Bertinetto et. al. (ECCV 2016 workshop paper)

---

> ### Author Response · Authors · 2020-11-14
> **About the reach of our contribution**
>
> We thank the reviewer for the valuable comments and insights. In the following we address specific concerns of the reviewer.
>
> **“That said, originality in terms of research contribution is limited. Siamese NNs have long been in use and recently revisited for variety of vision tasks such as face recognition [1, 2], object tracking [3], etc”**
>
> We agree that Siamese NNs is an architecture that has been previously studied, and other applications in different domains have been presented. However, we are the first to apply this type of architecture on the behavioral biometrics field. Our novelty relies on proposing a methodology that couples previous knowledge related to architecture, losses, sample generation and data structures, producing a well-performing model in a new domain. This behavioral domain presents its own difficulties inherent to the nature of biometric data, different from the challenges present in other tasks such as object tracking and face recognition tasks. Furthermore, we have shown that this approach is applicable and scalable to a real production case with thousands or millions of users, which has never been reported before.
>
> **“Comparisons with existing methods in the domain have not been included to the main paper (they are supposed to be in the appendix)."**
>
> We proved that a better performance is obtained when using a Siamese NNs with a few-shot interaction in comparison to previously studied approaches (SVM, RF, DT). Those comparisons were on the Appendix A, now, the most relevant have been moved to the main text.
>
> **"The aim perhaps of this paper is to present a baseline NN-based approach for user verification/ biometrics”**
>
> We showed that a single Siamese NN architecture abstracts the information from the input features so well, that two completely different sources (Desktop and Mobile) can be processed with the same Siamese NNs architecture. Moreover, we showed that our feature engineering process leads to an accurate latent space representation, which is useful for the biometric authentication use case.
>
> Given these points, the focus of our contribution is to combine one-shot learning and behavioral biometric authentication, a research intersection that remained unexplored in both ML and biometrics communities. Moreover, we propose a solution that overcomes implementation and scalability issues from previous approaches, while maintaining or improving performance in the real case. For instance, using only one model for all the users reduces the memory, training and monitoring requirements. Besides, compared with other approaches, we demand less interaction time per user to train and test. Those advantages arise from the few-shot nature of a Siamese NNs. We also contributed by confirming that, with the features proposed, there is an improvement of performance by using a triplet loss instead of a regular pair loss scheme. We would like to highlight that our main contribution is to show real world insights on the intersection between Siamese NNs and few-shot behavioral biometric authentication.

---

### Official Review · AnonReviewer1 · 2020-10-31
**Straightforward approach but limited ML contribution**

**Rating:** 4
**Confidence:** 3

**Review:**



Summary:


The paper proposes a siamese neural network to perform few-shot verification of user behaviour.

##########################################################################

Reasons for score:


Overall, I vote for rejection. I like the idea of applying the siamese paradigm to authentication however the paper offers limited novelty for a representation learning conference. There are zero baselines (no siamese) compared.


##########################################################################Pros:




1. The proposed architecture seems to be very accurate, as reported in the experimental section.

2. This paper provides comprehensive experiments, including both qualitative analysis and quantitative results across different datasets, to show the effectiveness of the proposed framework.


##########################################################################

Cons:


1. My major concern is the novelty of the siamese network. There seems to be no new component proposed that is tailored to biometric data.


2. I would suggest thinking about the contribution to the ML community. If the vanilla siamese network works like a charm, is the authentication problem solved?



##########################################################################

Questions during the rebuttal period:


Please acknowledge the contribution of the paper for an ML venue.

---

> ### Author Response · Authors · 2020-11-14
> **About the contribution to ICLR**
>
> We thank the reviewer for the valuable comments and insights. In the following we address specific concerns of the reviewer.
>
> **“I like the idea of applying the siamese paradigm to authentication however the paper offers limited novelty for a representation learning conference [...] My major concern is the novelty of the siamese network. [...]Please acknowledge the contribution of the paper for an ML venue.”**
> Regarding comments on novelty, even though Siamese architectures have been widely studied, their effectiveness on behavioral data has not been explored yet. Our contributions are focused on a practical study in the intersection of biometric data and Siamese architectures. We built suitable feature representations for human-device interactions from different behavioral sources (mouse, keyboard, mobile sensors), that enable the generation of a latent space effective for biometric authentication. Moreover, our proposed methodology for both web and mobile environments shows interesting insights about how biometric data and ML could be integrated in realistic scenarios, setting a precedent for future ML studies in the field. Furthermore, ICLR also encourages (1) studies showing applications on different fields and (2) research on implementation and scalability issues of current state-of-the art systems. Having said that, our work contributes to ML by proposing a one-shot ML framework for biometric data that has been proved in large scale in-the-wild scenarios and over different behavioral sources.
> An additional advantage of testing our approach on a large production dataset is that we were able to evaluate the impact of training with different user sample sizes (25K, 50K, 100K). This resulted in similar model performances even when the sample size was 4 times larger. This finding suggests that, for behavioral biometric data, there could be a limit to the representational power of the data in the latent space, this is an interesting insight for future research on Siamese neural networks applied to structured data.
>
> **“There are zero baselines (no Siamese) compared.”**
> Regarding a baseline, we showed that our chosen representation of human behavioral traits outperforms previous studies, even those using NN, on both web and mobile environments under practical scenarios with short interaction-time periods. On Appendix A, we compared the advantages of our approach against previously studied techniques found in literature. Notably, we are the only approach scalable to large deployments due to the application of one-shot techniques in behavioral biometrics. For the rebuttal phase, we have included numerical benchmarks with previous studies in the main document.

---

### Official Review · AnonReviewer5 · 2020-11-06
**Review of Paper 1323 (ICLR 2021)**

**Rating:** 9
**Confidence:** 4

**Review:**

Authors propose an approach to perform authentication based on users' behaviour (behavioral biometrics) on two different domains (web and mobile) by means of a siamese NN. The proposal is evaluated on three different dataset, collected ad-hoc for this work, showing a performance up to >0.9 acc. In the experimentation setup authors include different data splits, sequences lengths and training methods (pairs and triplets). The work shows the feasibility of using this type of approach for few-shot classification under real industry constraints.

------------------
Contributions

My recommendation for this manuscript is a strong accept. I believe this paper makes a relevant contribution to the field since it addresses a topic which is quite interesting for the audience, it is technically sound, covers different hypotheses and provides very useful insight to practitioners about the use of these approaches in real industry applications. The last point is the most interesting in my opinion, honestly.
The paper is well structured, with a clean format, very well written and easy to read.
Probably its weakest point is the novelty, since it addresses a well-known problem using a long-established NN architecture. However the combination of one-shot learning, siamese architecture and behavioral biometrics is something I personally have not seen before, so I consider it robust enough.

------------------
Recommendations

Although it is covered in the section 3.1, given the scope of the conference and its relevance for the reproducibility of the results, I’d ask authors to include a clear and detailed explanation of the feature vector employed. Some chart or graph where the reader can see easily how many features come from which source (sensor, keypad, gesture), how the derived features are computed and how movements are translated to features. Even better if the authors can include some examples.  Maybe as an appendix, but I really believe it’s important and a clear improvement for the manuscript.

I think it’d be interesting to add further details about the data splitting mechanism. I believe the paper needs at least to state clearly whether the train/test split is done by users (where a group of users are employed to perform inference and the rest as control test) or by sequences (where the data from each user is split into train and test). This is a relevant topic since the later has been demonstrated in previous HAR works to offer less robust and generalizable models. Please expand this point.

Figure 4 - Please specify clearly which sequence length is the one used for this experiment. It seems taken for granted it is 10 sec , since it’s the configuration offering best results, but please make it more clear to the reader.

From a personal perspective I think it’d be interesting to expand the work adding feature extraction mechanism using the state-of-art in time series modelling, this is, recurrent neural units or attention based mechanism. I’m really curious to see how that compares with the current engineered features.

------------------
Minor

Figure 2b - I’d recommend to scale the Accuracy axis since the series are hard to analyse (they are too close together)

Typo: “ Figure 2 shows the ROC curve and the FAR and FRR for the best setup in web environment” -> “Figure 3 shows the ROC curve and the FAR and FRR for the best setup in web environment”

Typo: “Figure 2 shows the ROC curve and the FAR and FRR for the best setup in mobile environment.” -> “ “Figure 3 shows the ROC curve and the FAR and FRR for the best setup in mobile environment.”

---

> ### Author Response · Authors · 2020-11-14
> **Adressing the recommendations**
>
> We thank the reviewer for the valuable comments and insights. In the following we address specific concerns of the reviewer. We have addressed, in the rebuttal version, the suggested modifications as follows.
>
> **"I believe the paper needs at least to state clearly whether the train/test split is done by users (where a group of users are employed to perform inference and the rest as control test) or by sequences (where the data from each user is split into train and test)."**
>
> Regarding the data split, we have expanded Section 4.2.1 making emphasis on our train/test split by users. We also added some remarks on the preference of splitting by users rather than history sequence.
>
>
> **"given the scope of the conference and its relevance for the reproducibility of the results, I’d ask authors to include a clear and detailed explanation of the feature vector employed. Some chart or graph where the reader can see easily how many features come from which source (sensor, keypad, gesture), how the derived features are computed and how movements are translated to features"**
>
> Moreover, we have added an appendix section (Appendix B - Feature Engineering Details) to expand the transformation from raw data to behavioral features.
>
> **"Figure 4 - Please specify clearly which sequence length is the one used for this experiment [...] Figure 2b - I’d recommend to scale the Accuracy axis since the series are hard to analyse (they are too close together)"**
>
> On the subject of paper figures, we have included references to time window length in Figure 4.  We also made some minor changes to Figure 2 related to improve visibility of series by scaling up the y axis.  Typos have been fixed too.

---

### Comment · AnonReviewer5 · 2020-11-10
**About the novelty of the work**

Dear colleagues,

I feel I should open the discussion phase on this paper since my review score is the most divergent from all reviewers

I can see the concerns of R3 and R1 about the limited novelty of the work regarding the algorithmic proposal, and that’s also what I stated as the main flaw of the manuscript. However, in my opinion this paper is still a clear accept.

The siamese architecture employed is indeed not novel at all, but I do think the value of this work comes from that. For me the main contribution of this paper is to show how an specific feature representation for biometric data can work in a real world user authentication use case. And for me it’s not just an incremental contribution (term which I do not really like). The combination of feature proposal, practical experimentation and actual real world insights can be of great interest for the audience. Just the dataset collected is a very significant contribution to the community.

I could accept that the contribution of the work is not enough if there would be not a significant advance from the current state of the art, but to the best of my knowledge (and after checking bibliography myself) I’ve not seen any other work that could make this paper as merely incremental.

Finally, I do not consider the paper is out of the scope of the conference. ICLR is indeed a proper venue to present advances in neural architectures but it is not its sole focus. For the case of this paper, whose main goal is to study the feasibility of its approach in real world scenarios, I could even see the use of a “simplistic” architecture as a pro rather than a con

---

### Decision · Program_Chairs · 2021-01-07
**Final Decision**

**Decision:**

Reject

**Comment:**

This paper received 3 reviews with mixed initial ratings: 9,4,5. The main concerns of R1 and R3, who gave unfavorable scores, included lack of novelty and hence limited value of this work for the ML community. At the same time, R5 strongly advocates for acceptance and mentions meaningful contributions in the context of the specific application, including the new dataset. In response to that, the authors submitted a new revision and provided detailed responses to each of the reviews separately, which did not change the position of the reviewers.
The AC agrees with R1 and R3 that, even though the biometrics-related contributions are relevant, the scope of this work is too narrow and application-driven for presentation in the main track of ICLR. As a result, the final recommendation is reject.